# Atrophic Gastritis and Autoimmunity: Results from a Prospective, Multicenter Study

**DOI:** 10.3390/diagnostics13091599

**Published:** 2023-04-30

**Authors:** Malgorzata Osmola, Caroline Hemont, Nicolas Chapelle, Marie-Anne Vibet, David Tougeron, Driffa Moussata, Dominique Lamarque, Edith Bigot-Corbel, Damien Masson, Justine Blin, Maxime Leroy, Regis Josien, Jean-François Mosnier, Jérôme Martin, Tamara Matysiak-Budnik

**Affiliations:** 1Department of Hematology, Transplantation and Internal Medicine, Medical University of Warsaw, 02-097 Warsaw, Poland; mosmola@wum.edu.pl; 2Department of Immunology, University Hospital of Nantes, 44093 Nantes, France; 3Institut des Maladies de l’Appareil Digestif (IMAD), Hepato-Gastroenterology & Digestive Oncology, University Hospital of Nantes, Hôtel Dieu, Place Alexis Ricordeau, CEDEX 1, 44093 Nantes, France; 4Institut National de la Santé et de la Recherche Médicale (INSERM) U1064 Centre de Recherche Translationnelle en Transplantation et Immunologie (CR2TI), 44093 Nantes, France; 5Faculty of Medicine, University of Nantes, 44300 Nantes, France; 6Department of Biostatistics, Centre Hospitalier Universitaire de Nantes, 44093 Nantes, France; 7Department of Hepato-Gastroenterology, Poitiers University Hospital, University of Poitiers, 86000 Poitiers, France; 8Department of Hepato-Gastroenterology, University Hospital of Tours, 37044 Tours, France; 9Department of Hepato-Gastroenterology, Ambroise-Paré Hospital, AP-HP, Paris Saclay University, University of Versailles Saint-Quentin-en-Yvelines, Institut National de la Santé et de la Recherche Médicale (INSERM), Infection and Inflammation, 91190 Paris, France; 10Department of Biochemistry, University Hospital of Nantes, 44093 Nantes, France; 11Institut National de la Santé et de la Recherche Médicale (INSERM) U1235 the Enteric Nervous System in Gut and Brain Disorders (TENS), 44300 Nantes, France; 12Department of Pathology, University Hospital of Nantes, 44093 Nantes, France

**Keywords:** autoimmune gastritis, chronic atrophic gastritis, autoimmunity, gastric cancer, *H. pylori*

## Abstract

Despite a global decrease, gastric cancer (GC) incidence appears to be increasing recently in young, particularly female, patients. The causal mechanism for this “new” type of GC is unknown, but a role for autoimmunity is suggested. A cascade of gastric precancerous lesions, beginning with chronic atrophic gastritis (CAG), precedes GC. To test the possible existence of autoimmunity in patients with CAG, we aimed to analyze the prevalence of several autoantibodies in patients with CAG as compared to control patients. Sera of 355 patients included in our previous prospective, multicenter study were tested for 19 autoantibodies (anti-nuclear antibodies, ANA, anti-parietal cell antibody, APCA, anti-intrinsic factor antibody, AIFA, and 16 myositis-associated antibodies). The results were compared between CAG patients (*n* = 154), including autoimmune gastritis patients (AIG, *n* = 45), non-autoimmune gastritis patients (NAIG, *n* = 109), and control patients (*n* = 201). ANA positivity was significantly higher in AIG than in NAIG or control patients (46.7%, 29%, and 27%, respectively, *p* = 0.04). Female gender was positively associated with ANA positivity (OR 0.51 (0.31–0.81), *p* = 0.005), while age and *H. pylori* infection status were not. Myositis-associated antibodies were found in 8.9% of AIG, 5.5% of NAIG, and 4.4% of control patients, without significant differences among the groups (*p* = 0.8). Higher APCA and AIFA positivity was confirmed in AIG, and was not associated with *H. pylori* infection, age, or gender in the multivariate analysis. ANA antibodies are significantly more prevalent in AIG than in control patients, but the clinical significance of this finding remains to be established. *H. pylori* infection does not affect autoantibody seropositivity (ANA, APCA, AIFA). The positivity of myositis-associated antibodies is not increased in patients with CAG as compared to control patients. Overall, our results do not support an overrepresentation of common autoantibodies in patients with CAG.

## 1. Introduction

With almost one million new cases every year, gastric cancer (GC) is the fifth most frequently diagnosed cancer and the third cause of cancer-related death worldwide [1]. According to the model of gastric carcinogenesis known as “Correa’s cascade” [2], GC is preceded by the sequential development of gastric precancerous lesions (GPL) (i.e., chronic atrophic gastritis (CAG), intestinal metaplasia (IM), and dysplasia), usually following a chronic infection with *Helicobacter pylori* (*H. pylori*) [2,3,4]. Less frequently, atrophic gastritis can result from an autoimmune reaction (autoimmune gastritis, AIG), which destroys gastric glands in the fundus [5,6,7]. In *H. pylori*-related gastritis, the lesions first appear in the antrum and eventually spread to the corpus [5,6,8,9]; in contrast, in AIG, the lesions are typically limited to the corpus (Figure 1a). 

Despite a global decrease in GC incidence over the last decades, recent epidemiological studies have shown a rising incidence in young, especially female, patients [10,11]. The causal mechanisms for this “new” type of GC have not been identified. However, a role for autoimmunity or changes in the microbiota has been proposed [11,12,13]. This is supported by recent studies suggesting an association between autoimmune conditions, such as dermatomyositis, Addison disease, and herpetiform dermatitis, and an increased risk of GC [14,15,16].

**Figure 1 diagnostics-13-01599-f001:**
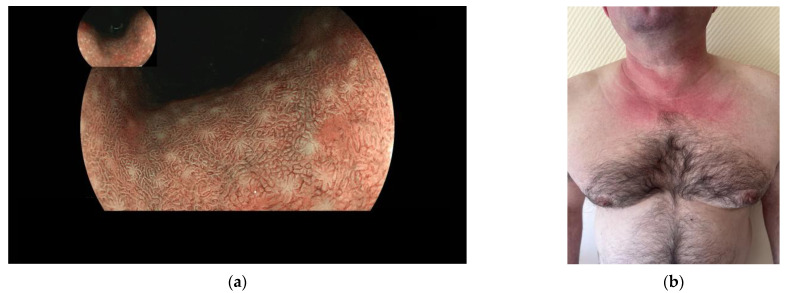
(**a**) Upper gastrointestinal endoscopy with virtual chromoendoscopy (BLI) showing intestinal metaplasia and gastric atrophy in the corpus in a patient with autoimmune gastritis. Photo from the private archive of Dr. Nicolas Chapelle. (**b**) A 45-year-old male patient with dermatomyositis presented with a skin rash and pruritus. Clinical examination revealed macular erythema over the sun-exposed parts of the anterior neck and upper chest, known as “V-sign”, a skin manifestation of dermatomyositis. Data from the literature indicate a strong association between dermatomyositis and GC [14,15]. Patient informed consent for the photo publication was obtained.

To test whether a possible overrepresentation of autoimmunity-associated autoantibodies in patients with CAG could exist, this study aimed to analyze the prevalence of routinely assessed autoantibodies in patients with CAG as compared to control patients. We tested 19 different autoantibodies, including anti-nuclear antibodies (ANA), anti-parietal cell antibody (APCA), anti-intrinsic factor antibody (AIFA), and 16 different myositis-associated antibodies. APCA and AIFA were included as “classical” AIG-associated antibodies [14], and ANA were included because of their presence in multiple autoimmune diseases [17]. The panel of myositis antibodies was selected according to the data from the literature indicating a strong association between dermatomyositis and GC [14,15], while its possible association with GPL has yet to be studied. The clinical picture of dermatomyositis is presented in Figure 1b.

## 2. Materials and Methods

### 2.1. Design of the Study

Serum samples collected from patients during our previous prospective, multicenter, cross-sectional study were analyzed. Out of 394 patients initially included in the study, 33 were excluded due to the absence of biopsies from two sites (corpus and antrum), 4 due to gastric adenocarcinoma at the initial examination, and 2 due to the lack of serum samples. Finally, 355 patients were included in the current study. Detailed descriptions of the study population, criteria for patient selection, endoscopy protocol used, blood sample collection, and histopathological evaluation of gastric biopsies were reported previously [18,19]. In brief, patients presented for upper endoscopy with gastric biopsies in four French University Hospitals between 2016 and 2019, and considered at increased risk of GC, were candidates for inclusion. Upper endoscopy with at least four gastric biopsies (two from the antrum and two from the corpus) was performed, and a fasting blood sample was obtained. The presence and intensity/distribution of GPL was evaluated with histopathological analysis of gastric biopsies according to the updated Sydney system [20]. The diagnosis of AIG was based on typical histology, including atrophic gastritis or intestinal metaplasia limited to the corpus with concomitant hyperplasia of enterochromaffin-like cells. Patients with CAG with typical histology were classified as NAIG. Other patients included in the study, with normal gastric mucosa or with non-atrophic gastritis on the histopathological examination, were classified as the control group. *H. pylori* status was assessed in all patients with histology and serology and was considered positive if at least one of the results was positive. 

### 2.2. Antibodies 

Nineteen autoantibodies, including ANA, APCA, AIFA, and 16 different myositis-associated antibodies were tested. APCA and AIFA were screened with fluorescence enzyme immunosorbent assay (FEIA) on an automated Phadia^TM^ 250 analyzer according to the supplier’s recommendations (Thermo Fisher Scientific Inc., Waltham, MA, USA). The cut-off values the manufacturer recommended are presented in Table 1.

ANA were screened with indirect immunofluorescence assay on HEp-2 cells (screening dilution 1:80) according to the supplier’s recommendations (Bio-Rad, Hercules, CA, USA). Positive sera were titrated with a 2-fold dilution up to a maximum of 1:2560. ANA results were classified as negative for dilution <1:80, equivocal for dilution 1:80, weakly positive for dilution 1:160, positive for dilution 1:320 or 1:640, and strongly positive for dilution ≥1:1280. 

Myositis autoantibodies were analyzed with Immunoblot assay (EUROLINE Myositis Profile; Euroimmun, Lübeck, Germany) according to the supplier’s recommendations. This immunoblot detected 12 myositis-specific autoantibodies (Mi-2α, Mi-2β, TIF1γ, MDA5, NXP2, SAE1, SRP, Jo-1, PL-7, PL-12, EJ, OJ) and 4 myositis-associated autoantibodies (Ku, PM-Scl100, PM-Scl-75, SSA-52). Immunoblot bands were analyzed with the EUROLineScan software (Euroimmun), allowing semi-quantitative determinations based on signal intensity (Table 1). 

### 2.3. Statistical Analysis

Differences between the groups with CAG (origin or location) versus controls were tested using Pearson’s chi-squared test for binary characteristics and the Student’s t or Fisher’s test for continuous characteristics. In order to identify characteristics that are more associated with ANA, AIFA, or APCA positivity, univariate and multivariate logistic regressions were carried out. Analyses were performed using R and R-studio. A significance level of *p* < 0.05 was adopted.

## 3. Results

### 3.1. Descriptive Analysis of the Study Population 

A comparison of demographic characteristics, *H. pylori* status, and autoantibody positivity between CAG and control patients is presented in Table 2. The data, according to the type of CAG (AIG or NAIG), are presented in Table 3. Patients were categorized into two major groups: patients with CAG (*n* = 154), and control patients (*n* = 201) including those with normal gastric mucosa or non-atrophic gastritis. Subsequently, within the CAG group, patients were classified into two sub-groups: autoimmune gastritis (AIG, *n* = 45) and non-autoimmune gastritis (NAIG, *n* = 109). In our cohort, patients in the CAG group were older than the control patients (mean age 61.5 ± 13.8 years vs. 56.4 ± 14.2 years, respectively, *p* < 0.001). Within the CAG group, NAIG patients were significantly older than control patients (62.5 ± 12.8 vs. 56.4 ± 14.2 years, respectively, with significance in post hoc analysis *p* < 0.001). There was no significant age difference between the AIG and control patients (58.9 ± 15.8 vs. 56.4 ± 14.2 years, *p* = 0.5). *H. pylori* infection was more frequent in the CAG than in the control group (27.3% vs. 15.4%, respectively, *p* = 0.006) and in NAIG as compared to AIG patients (33.9% vs. 11.1%, *p* = 0.02).

### 3.2. Autoantibodies

APCA and AIFA antibody positivity was overall significantly higher in the CAG group than in the control group (APCA 27% vs. 4%; AIFA 13.5% vs. 0, respectively, *p* < 0.001). Within the subgroups of CAG, APCA, and AIFA, antibody positivity was significantly higher in the AIG than in the NAIG and control groups (APCA: 73.3% vs. 7.5% vs. 4%, respectively, *p* < 0.001; AIFA: 40.5% vs. 2.8% vs. 0, respectively, *p* < 0.001, significant differences were noted between AIG and NAIG and AIG and controls, *p* < 0.001 for both antibodies), while there was no significant difference in APCA and AIFA seropositivity between the NAIG and control patients (Table 3). Although ANA positivity was not significantly different between CAG and the control group (*p* = 0.1), it was significantly higher in AIG than in NAIG or control patients (46.7%, 29%, and 27%, respectively, *p* = 0.03, a significant difference was present between AIG and control groups *p* = 0.04, and not between AIG and NAIG, *p* = 0.1) (Table 3 and Appendix A). 

Overall, there was no difference between the CAG and the control group with respect to myositis-associated antibodies positivity (Table 2). Myositis antibodies were found in 8.9%, 5.5%, and 4.4% of patients with AIG, NAIG, and in the control group, respectively, (*p* = 0.8) (Table 3). The antibody with the highest percentage of at least an equivocal result was PM75 (4.5% in the whole cohort). Beyond PM75, other myositis antibodies with at least equivocal results were detected only in less than 2% of the cohort (Appendix A).

### 3.3. Multivariate Analysis 

To look for other factors that could potentially affect the ANA, APCA, and AIFA seropositivity, we performed a multivariate analysis for the following factors: age, gender, and *H. pylori* infection. We found that the only factor influencing ANA positivity was female gender (OR 0.51 (0.31–0.81, *p* = 0.005)). Neither age nor *H. pylori* infection affected ANA seropositivity (Table 4). Whereas for APCA and AIFA, we found no factor affecting their positivity (Table 5). Considering that positivity for myositis antibodies was rare, it was not included in the multivariate analysis.

## 4. Discussion

It has been shown that different autoantibodies are more prevalent in patients with cancer, including GC [21,22], and that autoimmune diseases are associated with GC [14,15]. The aim of this study was thus to test the hypothesis that an increased prevalence of commonly assessed autoantibodies could be found already in patients with GPL, preceding the development of GC. Not surprisingly, APCA and AIFA positivity was more frequent in CAG than in control patients, explained by the high rate of seropositivity in patients with AIG [5]. No difference existed regarding ANA and myositis antibodies between CAG and controls, whereas ANA positivity was more frequent in AIG than in controls. To our knowledge, this is the first study investigating the ANA profile in a large group of patients with well-defined atrophic gastritis, particularly assessing the difference between the two types of chronic atrophic gastritis, autoimmune and *H. pylori*-induced. 

ANA positivity is detected in several autoimmune conditions, including systemic lupus erythematosus, systemic sclerosis, and Sjogren’s syndrome, but also in about 10% of the general population [23]. ANA are more prevalent in women and older individuals [24] and detected in around 30% of patients with malignancies [25]. In our study, seropositivity for ANA was detected in almost half of AIG patients (46.7%), which is a higher rate as compared to other studies, where seropositivity for ANA ranged between 17.4% in patients with AIG [26] to 19.1% in patients with *H. pylori*-negative CAG [27]. However, some of these studies were limited by a small sample size [26]. The higher ANA rate observed in our study may be related to the differences in methodology of ANA assessment, but also due to the high percentage of weakly positive results in our study (almost half of ANA positive results in AIG were weakly positive, Appendix A). Another possible explanation of high ANA seropositivity in AIG patients is the presence of concomitant autoimmune thyroiditis in patients with AIG, which might be associated with ANA seropositivity. In the literature, the seropositivity of ANA in autoimmune thyroiditis was described in 20–35% of patients [28,29]. We did not confirm the association between *H. pylori* infection and ANA positivity, as suggested by other studies [30]. The rate of ANA positivity in the control group in our study was also quite high (27%), but one third of the positive results were patients with weakly positive results (Appendix A). Our study confirms that high ANA might be partially attributed to a higher percentage of women in the AIG group. This is consistent with the data from the literature [24]. 

Another original investigation of our research was the assessment of myositis antibodies in CAG. Although we observed an overall low prevalence of myositis antibodies (5.3%), this rate appears higher than expected when compared to the general population (close to 1%) [31]. Consequently, firm conclusions cannot be drawn, given the lack of direct comparisons and different techniques used to analyze myositis antibodies. Interestingly, there was no association with a particular myositis antibody. The highest seropositivity was noted for PM75, which, together with PM100, are the antibodies characteristic for polymyositis, systemic sclerosis, and overlap syndromes [32,33]. Seropositivity for the PM75 antibody has low specificity which increases, in the case of double seropositivity for both PM75 and PM100, which was rare in our study. Other antibodies, including the most specific for dermatomyositis, associated with malignancy (NXP2 and TIF1g), remained low in our study population (0.3–0.6%) [34]. Thus, our results may instead suggest that dermatomyositis develops together with GC as a paraneoplastic syndrome and is not a causative factor [35].

Not surprisingly, APCA and AIFA were more prevalent in CAG than in the control group, but seropositivity of these antibodies is the hallmark of AIG and pernicious anemia [36,37]. On the other hand, APCA and AIFA positivity did not differ between the NAIG group and control patients. APCA is usually detected in 85–90% of AIG patients but may also be found in around 10% of the healthy population. AIFA is present in 35–60% of AIG cases and is highly specific for AIG [5,38]. APCA and AIFA can also be found in patients with other autoimmune diseases, such as celiac disease and diabetes mellitus type I [36,39]. 

The role of AIG as a precancerous condition is currently debated [40,41]. Some studies reported an increased GC risk in patients with AIFA [13], but recent studies found no association [42,43]. According to recent data, the increased GC risk reported in patients with AIG would be mainly related to the concomitant *H. pylori* infection [42,43]. Indeed, another important aspect is the role of *H. pylori* infection and its relationship to AIG. Some data suggest that *H. pylori* infection triggers AIG [44,45] and that *H. pylori* eradication may even lead to the regression of AIG [46]. However, the exact role of *H. pylori* in AIG has yet to be elucidated [5,36]. In the present study, *H. pylori* infection did not affect APCA and AIFA seropositivity, which is consistent with the data from the literature [47]. The association of *H. pylori* with the development of many autoimmune diseases (organ-specific and systemic) is evoked [48]. Conversely, the only autoimmune disease in which the role of *H. pylori* as a causative factor has been admitted is autoimmune thrombocytopenia [49]. 

Overall, our results do not support the initial hypothesis of the autoimmune response in patients with GPL beyond the known association with ACPA and AIFA. Nevertheless, they do not preclude that an autoimmune response may appear later in the gastric carcinogenesis.

Our study has several strengths, including its multicentric and prospective design. It is the first prospective study investigating the presence of autoantibodies, with an emphasis on myositis antibodies, in patients with well-defined CAG. The patients were divided according to the origin of gastritis (AIG and NAIG) to better understand the differences in autoimmunity in CAG. 

Our study also has some limitations. Firstly, the CAG group is relatively small. Even so, this condition is rare in regions with a low GC incidence, such as France (prevalence in Western Europe is around 3.2% [50], compared to >20% in Southeast Asia and South America [51]). Secondly, we did not adjust the antibody’s level according to information from past medical history, such as the history of autoimmune diseases, which is a major drawback, but the initial study design did not imply the collection of these data from the patients. Moreover, the median age in our cohort is above 50 years. Therefore, the higher level of antibodies may be related to age, even though multivariate analysis did not confirm the influence of age on antibody seropositivity. 

## 5. Conclusions

Overall, our results do not support the association between the presence of common autoantibodies, particularly myositis-associated antibodies, and GPL, except for an expected overrepresentation of APCA and AIFA in AIG. Interestingly, ANA appear more prevalent in AIG than in control patients, and the significance of this finding, both on pathophysiological and diagnostic levels, deserves further investigation. Additionally, *H. pylori* infection does not appear to affect the autoantibody positivity (ANA, APCA, AIFA).

## Figures and Tables

**Table 1 diagnostics-13-01599-t001:** Antibodies and the cut-off values.

Antibody	Negative	Equivocal	Positive
APCA, AIFA [U/mL]	<7	7–10	>10
ANA	<1:80	1:80	≥1:160
Myositis-associated antibodies	≤10	>10	>25

APCA, anti-parietal cell antibody; AIFA, anti-intrinsic factor antibody; ANA, anti-nuclear antibodies; myositis-associated antibodies including Mi-2α, Mi-2β, TIF1γ, MDA5, NXP2, SAE1, SRP, Jo-1, PL-7, PL-12, EJ, OJ, Ku, PM-Scl100, PM-Scl-75, SSA-52 were assessed.

**Table 2 diagnostics-13-01599-t002:** Comparison of patient characteristics, autoantibody seropositivity, and *H. pylori* status in chronic atrophic gastritis and control patients.

Parameter	CAG (*n* = 154)	Control (*n* = 201)	*p*-Value	Total (*n* = 355)
Age (year) mean (±SD)	61.5 (±13.8)	56.4 (±14.2)	<0.001	58.6 (±14.2)
Range (year)	22–89	18–82		18–89
Sex			0.09	
Female *n* (%)	76 (49.4)	117 (58.2)		193 (54.4)
Male *n* (%)	78 (50.6)	84 (41.8)		162 (45.6)
*H. pylori* status			0.006	
Histology positive *n* (%)	25 (16.2)	22 (10.9)		47 (13.2)
Serology positive *n* (%)	35 (22.7)	27 (13.4)		62 (17.5)
Any *H. pylori* positive *n* (%)	42 (27.3)	31 (15.4)		73 (20.6)
APCA *n* (%)	41 (27.0)	8 (4.0)	<0.001	49 (13.9)
AIFA *n* (%)	20 (13.5)	0	<0.001	20 (5.8)
ANA *n* (%)	52 (34.2)	54 (27.0)	0.1	106 (30.1)
Myositis-associated antibodies			0.6	
At least one antibody equivocal or positive *n* (%)	22 (14.5)	26 (12.9)		59 (13.8)
At least one positive antibody *n* (%)	9 (5.9)	9 (4.4)		19 (5.3)

CAG, chronic atrophic gastritis; APCA, anti-parietal cell antibody; AIFA, anti-intrinsic factor antibody. Cut-off values for APCA and AIFA, negative: <7 U/mL, equivocal: 7–10 U/mL, positive: >10 U/mL. Values qualified as positive for APCA and AIFA were with cut-off >10 U/mL. ANA, anti-nuclear antibodies; ANA results: negative dilution <1:80, equivocal 1:80, positive ≥1:160. Values qualified as positive for ANA were ≥1:160. Myositis-associated antibodies seropositivity, equivocal > 10; positive >25; myositis antibodies included Mi-2α, Mi-2β, TIF1γ, MDA5, NXP2, SAE1, SRP, Jo-1, PL-7, PL-12, EJ, OJ, Ku, PM-Scl100, PM-Scl-75, SSA-52; *H. pylori, Helicobacter pylori*. Values are presented as n (%), mean (±SD). Pearson’s chi-squared test or Linear Model ANOVA was used for statistical analysis, and a significance level of *p* < 0.05 was adopted.

**Table 3 diagnostics-13-01599-t003:** Comparison of patients’ characteristics, *H. pylori* status, and antibody seropositivity among the patients with autoimmune gastritis, with non-autoimmune gastritis, and control patients.

Parameter	AIG (*n* = 45)	NAIG (*n* = 109)	Control (*n* = 201)	*p*-Value	Total (*n* = 355)
Age (year) mean (±SD)	58.9 (±15.7)	62.5 (±12.8)	56.4 (±14.2)	0.001	58.6 (±14.2)
Range (year)	23–89	22–87	18–82		18–89
Sex				0.059	
Female *n* (%)	27 (60.0)	49 (45.0)	117 (58.2)		193 (54.4)
Male *n* (%)	18 (40.0)	60 (55.0)	84 (41.8)		162 (45.6)
*H. pylori* status				<0.001	
Histology positive *n* (%)	0	25 (22.9)	22 (10.9)		47 (13.2)
Serology positive *n* (%)	5 (11.1)	30 (27.5)	27 (13.4)		62 (17.5)
Any *H. pylori* positive *n* (%)	5 (11.1)	37 (33.9)	31 (15.4)		73 (20.6)
APCA *n* (%)	33 (73.3)	8 (7.5)	8 (4.0)	<0.001	49 (13.9)
AIFA *n* (%)	17 (40.5)	3 (2.8)	0	<0.001	20 (5.8)
ANA *n* (%)	21 (46.7)	31 (29.0)	54 (27.0)	0.03	106 (30.1)
Myositis antibodies				0.8	
At least one antibody equivocal or positive *n* (%)	7 (14.3)	15 (15.6)	26 (12.9)		59 (13.8)
At least one positive antibody *n* (%)	4 (8.9)	6 (5.5)	9 (4.4)		19 (5.3)

AIG, autoimmune gastritis; NAIG, non-autoimmune gastritis; APCA, anti-parietal cell antibody; AIFA, anti-intrinsic factor antibody. Cut-off values for APCA and AIFA, negative: <7 U/mL, equivocal: 7–10 U/mL, positive: >10 U/mL. Values qualified as positive for APCA and AIFA with cut-off >10 U/mL. ANA, anti-nuclear antibodies; ANA results: negative dilution <1:80, equivocal 1:80, positive ≥1:160. Values qualified as positive for ANA were ≥1:160. Myositis-associated antibodies seropositivity, equivocal >10; positive >25; myositis antibodies included Mi-2α, Mi-2β, TIF1γ, MDA5, NXP2, SAE1, SRP, Jo-1, PL-7, PL-12, EJ, OJ, Ku, PM-Scl100, PM-Scl-75, SSA-52; *H. pylori, Helicobacter pylori.* Values are presented as n (%) or mean (± SD). Pearson’s chi-squared test or Linear Model ANOVA was used for statistical analysis; a significance level of *p* < 0.05 was adopted.

**Table 4 diagnostics-13-01599-t004:** Multivariate analysis for ANA.

Parameter		ANA Negative	ANA Positive	OR (Univariate)	OR (Multivariate)
Age *n* (%)	≤50	70 (72.2)	27 (27.8)		
	>50	176 (69.0)	79 (31.0)	1.16 (0.70–1.97, *p* = 0.5)	1.23 (0.73–2.11, *p* = 0.4)
Sex *n* (%)	Female	122 (63.5)	70 (36.5)		
	Male	124 (77.5)	36 (22.5)	0.51 (0.31–0.81, *p* = 0.005)	0.50 (0.31–0.80, *p* = 0.004)
*H. Pylori n* (%)	Negative	199 (71.1)	81 (28.9)		
	Positive	47 (65.3)	25 (34.7)	1.31 (0.75–2.25, *p* = 0.3)	1.31 (0.74–2.27, *p* = 0.3)

ANA, anti-nuclear antibodies; ANA results: negative, dilution <1:80, positive, ≥1:16; *H. pylori, Helicobacter pylori*. OR, odds ratio (95% confidence interval). Values are presented as n (%). The chi-square test was used for statistical analysis.

**Table 5 diagnostics-13-01599-t005:** Multivariate analysis for APCA and AIFA.

Parameter		APCA Negative	APCA Positive	OR (Univariate)	OR (Multivariate)	AIFA Negative	AIFA Positive	OR (Univariate)	OR (Multivariate)
Age *n* (%)	≤50	80 (82.5)	17 (17.5)			87 (90.6)	9 (9.4)		
	>50	223 (87.5)	32 (12.5)	0.68 (0.36–1.31, *p* = 0.2)	0.69 (0.37–1.34, *p* = 0.3)	240 (95.6)	11 (4.4)	0.44 (0.18–1.13, *p* = 0.08)	0.46 (0.18–1.12, *p* = 0.09)
Sex *n* (%)	Female	163 (84.9)	29 (15.1)			176 (93.6)	12 (6.4)		
	Male	140 (87.5)	20 (12.5)	0.80 (0.43–1.47, *p* = 0.5)	0.83 (0.44–1.52, *p* = 0.5)	151 (95.0)	8 (5.0)	0.78 (0.30–1.93, *p* = 0.6)	0.85 (0.34–2.09, *p* = 0.7)
*H. Pylori n* (%)	Neg.	239 (85.4)	41 (14.6)			258 (92.8)	20 (7.2)		
	Pos.	64 (88.9)	8 (11.1)	0.73 (0.30–1.56, *p* = 0.4)	0.74 (0.31–1.58, *p* = 0.5)	69 (100.0)	0	-	0.09 (0.006–1.5, *p* = 0.09)

APCA, anti-parietal cell antibody; AIFA, anti-intrinsic factor antibody. Cut-off values for APCA and AIFA, negative: <7 U/mL, equivocal: 7–10 U/mL, positive: >10 U/mL.Values qualified as positive for APCA were with cut-off >10 U/mL; *H. pylori, Helicobacter pylori;* Neg., negative; Pos., positive. OR, odds ratio (95% confidence interval). Values are presented as n (%). The chi-square test was used for statistical analysis.

## Data Availability

The data presented in this study are available on request from the corresponding author. The data are not publicly available due to protection of patients’ privacy.

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
