# Peer review of "Atrophic Gastritis and Autoimmunity: Results from a Prospective, Multicenter Study"

_diagnostics, 2023, doi:10.3390/diagnostics13091599_

Round 1

Reviewer 1 Report

After a careful reading, this manuscript has no objective errors. Although this analysis is relatively simple, it does bring novel insights. The methods are clearly described, and the conclusions and discussion harmonize with the obtained results. The authors do not make doubtful inferences and are fair with their study limitations. In fact, they explained my major concerns in their " limitation" section. Based on their arguments, the limitations are acceptable.
The only thing is that the manuscript needs minor grammar checks, especially regarding punctuation.

Author Response

Dear Reviewer, 

Thank you so much for your time in reviewing our article. We are glad that you appreciate the methodology, which we consider a strong point of our study. We tried to explain point-by-point our study's limitations. The English language was checked by a native speaker before the resubmission, as demanded. We hope that this article will be a valuable addition to the field, 

With our best regards, 

Reviewer 2 Report

I have read your article entitled "Atrophic gastritis and autoimmunity: Results from a prospective, multicenter study" with great interest. The study investigated the relationship between atrophic gastritis and autoimmune disorders, which is an important topic that deserves attention.

Overall, I found your study to be well designed and informative. The sample size was large and the multicenter approach added to the credibility of the results.

However, there are a few areas where I think more information would be beneficial.

The aims of the study presented in the Introduction and mainly in the Abstract are not informative, and do not correspond to what was performed in the study.

It is not clear how the control groups were selected. Note that this is crucial information, and the lack of information lead to suspicion for a risk for selection bias.

Since the authors are investigating the role of autoimmune antibodies, a previous history of autoimmune conditions should be clarified as baseline characteristics of the included patients.

In Table 3, use sentence case for “Age Mean”.

Overall, I found the study to be a valuable contribution to the field, and I appreciate the careful attention to detail that was evident throughout the article.

Author Response

We would like to thank you  for your valuable comments and suggestions that have allowed us to improve the manuscript,

Please, find below our point-by-point responses to your comments,

Comment: The aims of the study presented in the Introduction and mainly in the Abstract are not informative, and do not correspond to what was performed in the study.

Response: We modified the abstract and introduction accordingly

Abstract: “To test the possible existence of autoimmunity in patients with CAG, we aimed to analyze the prevalence of several autoantibodies in patients with CAG as compared to control patients.”

Introduction: “The panel of myositis antibodies was selected according to the data from the literature indicating a strong association between dermatomyositis and GC, while its possible association with GPL has yet to be studied. Clinical picture of dermatomyositis is presented on Figure 1b."

Comment: It is not clear how the control groups were selected. Note that this is crucial information, and the lack of information lead to suspicion for a risk for selection bias.

Response: The definition of the control group was based on histology of the gastric mucosa, evaluated according to the Sydney system, and defined as either a normal mucosa or non-atrophic gastritis. This definition has been now more clearly described: “Patients with a normal mucosa or with a non-atrophic gastritis based on the histopathological examination, were classified as the control group.”

Comment: Since the authors are investigating the role of autoimmune antibodies, a previous history of autoimmune conditions should be clarified as baseline characteristics of the included patients.

Response: This is a very important point. Indeed, this information was not collected and it is one of the drawbacks of the study, clearly indicated in the discussion section:  “we did not adjust the antibody's level to the information from past medical history, such as the history of autoimmune diseases, which is a major drawback, but the initial study design did not imply collection of such data from the patients. Additionally, collecting such data after many years from the inclusion to the study, could have been a confounding factor, so it was not conducted.”

Comment: In Table 3, use sentence case for “Age Mean”.

Response:  We addressed the “Age Mean” comment in the manuscript.

English language was checked by a native speaker before the submission, as demanded.

We hope that you will find our manuscript improved!

With our best regards,